# Inter- and intra-observer reliability and agreement of O₂Pulse inflection during cardiopulmonary exercise testing: A comparison of subjective and novel objective methodology

Thomas Nickolay[1,2]*, Gordon McGregor[3,4,5], Richard Powell[3,4], Brian Begg[6,7], Stefan Birkett[8], Simon Nichols[9], Stuart Ennis[3,5], Prithwish Banerjee[4,5,10], Rob Shave[11], James Metcalfe[2], Angela Hoye[1], Lee Ingle[2]

1 Hull York Medical School, University of Hull, Hull, United Kingdom, 2 School of Sport, Exercise & Rehabilitation Science, Faculty of Health Sciences, University of Hull, Kingston-Upon-Hull, United Kingdom, 3 Department of Cardiopulmonary Rehabilitation, Centre for Exercise & Health, University Hospitals Coventry & Warwickshire NHS Trust, Coventry, United Kingdom, 4 Centre for Physical Activity, Sport & Exercise Sciences, Coventry University, Coventry, United Kingdom, 5 Warwick Clinical Trials Unit, Warwick Medical School, University of Warwick, Coventry, United Kingdom, 6 Cardiff Centre for Exercise & Health, Cardiff Metropolitan University, Cardiff, Wales, United Kingdom, 7 Aneurin Bevan University Health Board, Gwent, Wales, United Kingdom, 8 Department of Sport and Exercise Sciences, Manchester Metropolitan University, Manchester, United Kingdom, 9 Nursing, Midwifery, and Paramedic Practice, Robert Gordon University, Aberdeen, United Kingdom, 10 Department of Cardiology, University Hospitals Coventry & Warwickshire NHS Trust, Coventry, United Kingdom, 11 Centre for Heart Lung and Vascular Health, University of British Columbia—Okanagan, Kelowna, Canada

* t.nickolay@hull.ac.uk

**Data Availability Statement:** All Master files are available from the Open Science Framework database https://osf.io/d4pv3/.

## Abstract

Cardiopulmonary exercise testing (CPET) is the 'gold standard' method for evaluating functional capacity, with oxygen pulse (O₂Pulse) inflections serving as a potential indicator of myocardial ischaemia. However, the reliability and agreement of identifying these inflections have not been thoroughly investigated. This study aimed to assess the inter- and intra-observer reliability and agreement of a subjective quantification method for identifying O₂Pulse inflections during CPET, and to propose a more robust and objective novel algorithm as an alternative methodology. A retrospective analysis was conducted using baseline data from the HIIT or MISS UK trial. The O₂Pulse curves were visually inspected by two independent examiners, and compared against an objective algorithm. Fleiss' Kappa was used to determine the reliability of agreement between the three groups of observations. The results showed almost perfect agreement between the algorithm and both examiners, with a Fleiss' Kappa statistic of 0.89. The algorithm also demonstrated excellent inter-rater reliability (ICC) when compared to both examiners (0.92–0.98). However, a significant level ($P \leq 0.05$) of systematic bias was observed in Bland-Altman analysis for comparisons involving the novice examiner. In conclusion, this study provides evidence for the reliability of both subjective and novel objective methods for identifying inflections in O₂Pulse during CPET. These findings suggest that further research into the clinical significance of O₂Pulse

**Funding:** The author(s) received no specific funding for this work.

**Competing interests:** The authors have declared that no competing interests exist.

inflections is warranted, and that the adoption of a novel objective means of quantification may be preferable to ensure equality of outcome for patients.

## Introduction

Cardiopulmonary exercise testing (CPET) allows for the non-invasive, objective quantification of cardiopulmonary fitness, and is thus held as the 'gold standard' methodology for evaluating functional capacity [1, 2]. In contrast to more traditional assessments, such as ECG stress testing and the 6-minute walk test, CPET makes it possible to determine the potential pathophysiological mechanisms underlying exercise intolerance [2–4].

The utility of CPET as a diagnostic and prognostic tool in the evaluation of patients with coronary artery disease (CAD) and heart failure has received some attention [1–5]. In particular, an early plateau or inflection in the normal linear progression of oxygen pulse ($O_2$Pulse) and oxygen consumption ($VO_2$) despite an increasing work-rate (WR) are suggested to be indicative of inducible and reversable ischaemia [2, 4, 6–9]. In principle, $O_2$Pulse reflects left ventricular stroke volume (SV) (and arteriovenous oxygen difference) [1, 4]. Consequently, a plateau or inflection in $O_2$Pulse, despite increasing WR suggests a pathological impairment of stroke volume, possibly caused by myocardial ischaemia [1, 2, 4].

To the best of our knowledge the reliability and agreement of $O_2$Pulse inflections have not been previously investigated. However, we have shown in a healthy cohort, the minimal detectable change (MDC) for 15-second time-averaged and filtered $O_2$Pulse, measured between 50 and 100% of peak work rate is 2.2 mL.beat, and 1.6 mL.beat respectively.

If this level of agreement remains consistent for $O_2$Pulse at the point of inflection, and this morphology represents stable pathophysiological limitations in CAD patients, it may be very useful to clinicians. For example, it could provide a marker with which to track the progression and severity of dysfunction, without the need for repeated exposure to radiation or invasive procedures. Moreover, in rehabilitation settings it may provide a threshold value from which personalised exercise prescriptions could be developed.

However, to date the literature surrounding inflections in $O_2$Pulse typically classifies them categorically, as 'normal' or 'abnormal'. This system of classification does not quantify the position of inflection, for example the work rate or heart rate at which $O_2$Pulse deviates from normality [1, 10–15]. Categorisation of $O_2$Pulse morphology is usually performed in one of two ways, which we refer to as 'visual categorisation', and secondly, 'categorisation by regression'. In visual categorisation, one or more observers scrutinise the $O_2$Pulse curve to identify where the curve begins to deviate from a linear increase, usually referred to as a plateau or inflection. This point can sometimes be the sole focus of the investigation. However, in other instances, observers may be required to further categorise the curve as 'normal', 'probably normal', 'probably abnormal', or 'abnormal', depending on its characteristics [10]. Alternatively, 'categorisation by regression' involves a quantified mathematical approach. Investigators select a specific point along the curve, such as two minutes before the cessation of exercise, and calculate the slope of the curve (slope A) from exercise beginning to this point using linear regression. This slope is then compared to the regression slope of the curve for the final two minutes (slope B) to quantify proportional change [16]. Based on this comparison, the curve may be further categorised as 'normal augmentation', 'flat throughout', plateau in late exercise', and 'decline in late exercise' (inflection) [16].

The inter-rater agreement when categorising $O_2$Pulse curves as 'normal', 'probably normal', 'probably abnormal', 'definitely abnormal' has been reported by De Lorenzo and colleagues [10] to be κC = 0.65 (95% CI = 0.39–0.66). Efforts have been made by Chuang et al [15] to remove the subjectivity from categorisation by comparing an algorithmic approach to the consensus of two examiners. The resulting Kappa values were κC = 0.86 and 0.69 respectively for the conditions (normal) plateau and decrease.

To date there appears to have been no effort made to subjectively quantify the position of $O_2$Pulse inflections. This is perhaps due to their identification being influenced by a multitude of factors, such as the experience and opinion of the individual interpreting the data, the method of data processing (time averaged versus data point averaging), and data presentation (axis size and aspect ratio). However, this form of data interpretation is not without precedent, the first ventilatory threshold (VT1) during CPET is often identified in much the same way, through the modified V-slope method [17]. Harwood et al. [18] investigated the agreement of CPET parameters in patients with abdominal aortic aneurysms, utilising modified V-slope method to identify VT1. The intraclass correlation coefficient (ICC) (two-way mixed) was used to measure reliability. For intra-rater reliability, the ICC was 0.834 (95% CI 0.215,0.975; $P = 0.010$) on the motorised treadmill and r = 0.959 (95% CI 0.741,0.994; $P = 0.000$) on the cycle ergometer. For inter-rater reliability, the ICC was r = 0.983 (95% CI 0.785,0.999; $P = 0.002$) on the motorised treadmill and r = 0.905 (95% CI 0.508,0.986; $P = 0.003$) on the cycle ergometer.

$O_2$Pulse inflections may have the potential to be used to prescribe exercise intensity and monitor progression in much the same way VT1 is currently used. However, we must first establish the inter- and intra-rater variability of identification, and potentially provide a robust objective means of identification.

To the best of our knowledge there are no published data relating to the inter-, or intra-observer reliability and agreement of the subjective quantification of $O_2$Pulse inflections. Therefore, the primary aim of this research is to determine the inter- and intra-observer reliability and agreement of the subjective quantification method. The secondary aim is to establish a suitable objective alternative methodology that provides zero intra- and inter- observer variability.

## Methods

This was a retrospective baseline analysis of the HIIT or MISS UK trial [18]. The HIIT or MISS trial was a multicentre randomised controlled trial recruiting (1st September 2016 to 13th March 2020) CAD patients referred for exercise-based cardiac rehabilitation (CR) in the UK. Ethical approval for the protocol was provided by the NHS Health Research Authority, East Midlands–Leicester South Research Ethics Committee (16/EM/0079), with patients providing written informed consent prior to enrollment. Detailed methodology of the trial procedures are available elsewhere [19], in short, patients performed a baseline CPET on cycle ergometer following a standard ramp incremental protocol [20]. Fully anonymised data were accessed for the purpose of this analysis between 29th January 2021 and 28th September 2023. Raw ventilatory gas exchange data were exported as 15-second averaged.csv files and used to generate $O_2$Pulse curves (x-axis = work rate; y-axis = $O_2$Pulse).

Curves were then visually inspected by two independent examiners, each blinded to the interpretation of the other. Both examiners were clinical exercise physiologists with experience interpreting CPET. However, one examiner had substantially more experience with $O_2$Pulse morphology (>6 years) and inflections. This examiner is subsequently referred to as 'experienced' whilst the other is termed 'novice' (<1 year). Each examiner viewed all available

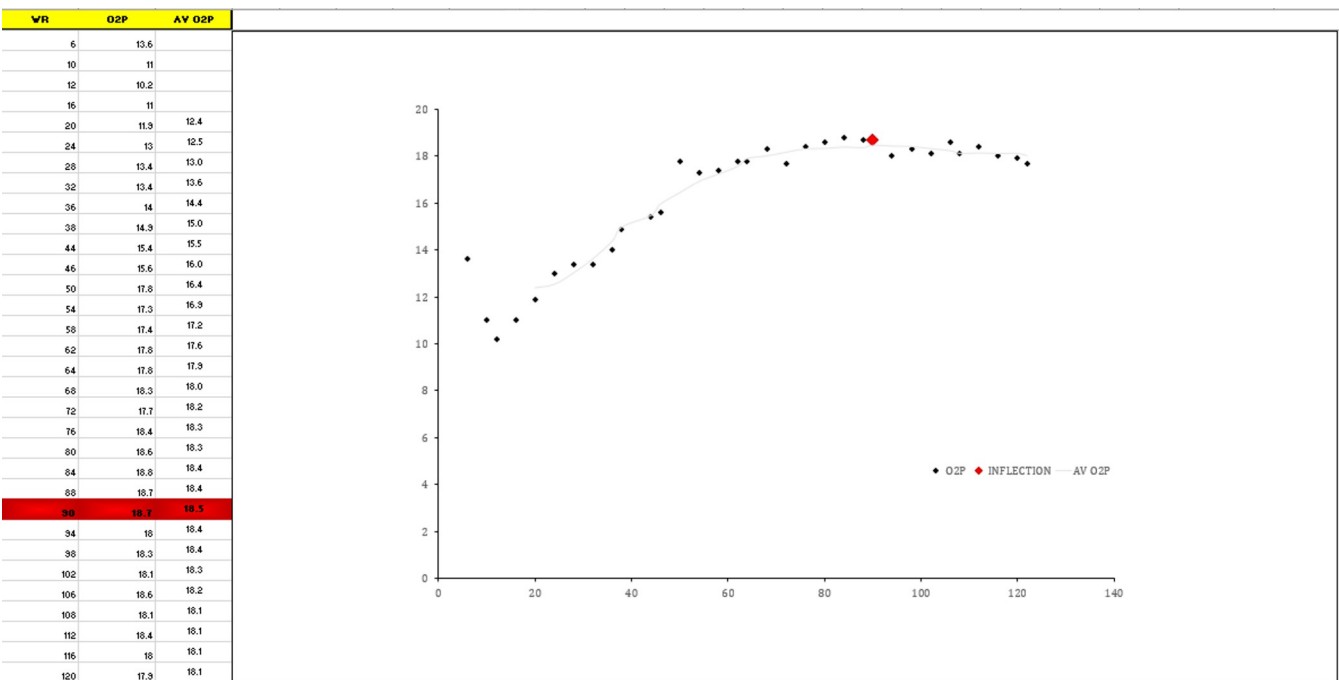

**Fig 1. Example of the algorithm function as an excel template.**

$O_2$Pulse curves, categorising each as 'yes', to indicate the perceived presence of inflection or plateau, or 'no' to indicate the normal linear progression of $O_2$Pulse. For each curve categorised as 'yes' the examiner would then quantify the threshold for inflection, identifying the exact point in the plot they believed represented a departure from normality. The 'experienced' examiner revisited the 'yes' curves at a later date (7–14 days) to re-quantify the inflection threshold. All subjective observations were then compared against an objective algorithm to compare categorisation and quantified threshold placement.

The algorithm was developed around the principle that a linear, or curvilinear increase in $O_2$Pulse was the expected normal response. The algorithm functions on 15-second time averaged data. To further reduce data noise without having to identify and correct for individual out-lying data points, we applied a 9-point moving average filter. This process involved replacing each data point with the average of the 9 data points centred around it, which included the 4 preceding points, the point itself, and the 4 subsequent points. From this processed data the algorithm simply identified the first occurrence of the series peak value using conditional formatting. If this peak value occurred $\geq$ 6 data points prior to the end of the test, the row was highlighted as a departure from normal linear or curvilinear increase, and the point was plotted on the embedded figure for visual inspection (Fig 1). Using the algorithm template, this requires the 15-second averaged work rate and $O_2$Pulse data to be copied and pasted into columns A and B.

The additional criteria specified for accepting inflections in $O_2$Pulse, either via subjective observer or algorithm were as follows:

The point of inflection should occur $\geq$ 6 data points (90 seconds) prior to the end of the test.

The point of inflection in $O_2$Pulse should coincide with a reduction in the $\Delta VO_2/\Delta WR$ slope of $\geq$ 10%.

The patient must not have achieved $\geq$ 90% predicted $VO_{2peak}$.

These criteria are derived from the original findings of Belardinelli and colleagues [7, 8].

### Statistical analysis

Statistical analysis was performed in RStudio version 4.2.2 using the R programming language and packages "readxl", "irr", "epiR", and "BlandAltmanLeh" (Integrated Development for R. PBC, Boston, MA, USA). The categorisations of each observer (coded as "Yes" or "No" to indicate the perceived presence or absence of an inflection or plateau) were compared against that of the objective algorithm to establish whether the algorithm could adequately categorise inflections. Fleiss' Kappa ($\kappa$F) was used to determine the reliability of agreement between the three groups of observations. Kappa statistics were interpreted in accordance with the suggestions of Landis and Koch [21] with values <0.00, 0.00–0.20, 0.21–0.40. 0.41–0.60, 0.61–0.80 and 0.81–1.00 indicating poor, slight, fair, moderate, substantial and almost perfect respectively. In order to provide 95% confidence intervals around the Kappa value we performed 1000 bootstrap resamples with replacement from the original dataset. The algorithm was also compared against the consensus of both subjective examiners to determine its sensitivity and specificity as well as both positive and negative predictive values.

If all three observations were in agreement that an inflection had occurred, the threshold for inflection, expressed as heart rate and work rate were visually compared with Bland-Altman plots. In these instances, we compared experienced to novice, experienced to experienced (time), experienced to algorithm, and novice to algorithm. The intra-rater reliability was compared with a two-way random effects (2,1) ICC for absolute agreement and reported with standard error of measure (SEM) and minimal detectable change (MDC) values. The inter-rater reliability were compared using two-way mixed effects (3,1) model ICCs for absolute agreement [22]. ICC outputs were interpreted based upon the recommendations of Koo and Li [22] with values <0.5, 0.5–0.75, 0.75–0.9 and >0.9 indicating poor, moderate, good and excellent reliability respectively. Statistical significance was accepted $P \leq 0.05$.

## Results

In total 272 baseline CPET data in patients with CAD were analysed. The results of the analyses are presented in two parts: first, the inter-observer agreement of the subjective categorisation method versus the objective algorithm, and second, the evaluation of the proposed objective algorithms for quantifying thresholds in $O_2$Pulse.

### Inter-observer agreement

The computed Fleiss' Kappa statistic for all raters was $\kappa$F = 0.89 with a bootstrapped 95% confidence interval of 0.83–0.93. The corresponding z-score was 25.5 with a $P < 0.0001$. At least two raters were in agreement across all 272 files, with all three raters in agreement on 260 occasions (95.6%) The comparison of each interpreters' analysis is summarised in Table 1.

**Table 1. Comparison of sensitivity, specificity, positive predictive value, and negative predictive value across different rater comparisons.**

| Comparison | Sensitivity (95% CI) | Specificity (95% CI) | Positive Predictive Value (95% CI) | Negative Predictive Value (95% CI) |
|---|---|---|---|---|
| Algorithm Vs. Experienced | 0.93 (0.82–0.99) | 0.98 (0.96–1.00) | 0.91 (0.79–0.98) | 0.99 (0.96–1.00) |
| Algorithm Vs. Novice | 0.89 (0.76–0.96) | 0.99 (0.96–1.00) | 0.93 (0.81–0.99) | 0.98 (0.95–0.99) |
| Experienced Vs. Novice | 0.93 (0.81–0.99) | 0.97 (0.94–0.99) | 0.87 (0.74–0.95) | 0.99 (0.96–1.00) |
| Algorithm Vs. Consensus | 0.82 (0.68–0.92) | 0.99 (0.96–1.00) | 0.92 (0.80–0.98) | 0.97 (0.93–0.98) |

95% CI = 95% Confidence Interval

## Evaluation of the objective algorithm

In instances where all three observations agreed that an inflection had occurred (n = 37; 13.6%), the threshold for inflection, expressed as heart rate and work rate, were compared using Bland Altman plots (Figs 2 and 3). Values derived from or associated with Bland-Altman analysis, along with ICC values, are reported in Table 2.

Excellent reliability was recorded for all ICC, with the highest heart rate values (0.97) occurring in both the experienced and novice versus algorithm comparisons. The highest work rate ICC occurred in the algorithm versus experienced comparison (0.98). The intra-rater reliability for work rate (0.95) was accompanied by SEM (%SEM) and MDC (%MDC) values of 11.2 (11.1%) and 15.53 (15.4%) respectively. Whilst the intra-rater heart rate (0.95) SEM, and MDC values or 7.13 (6.8%) and 9.88 (9.4%).

Although ICC for the novice comparisons involving the novice examiner were excellent, the 95% CI for all of these readings were consistently broader than those involving the

**Fig 2. Bland-Altman plots comparing agreement across subjective and objective inflection identification for heart rate.**

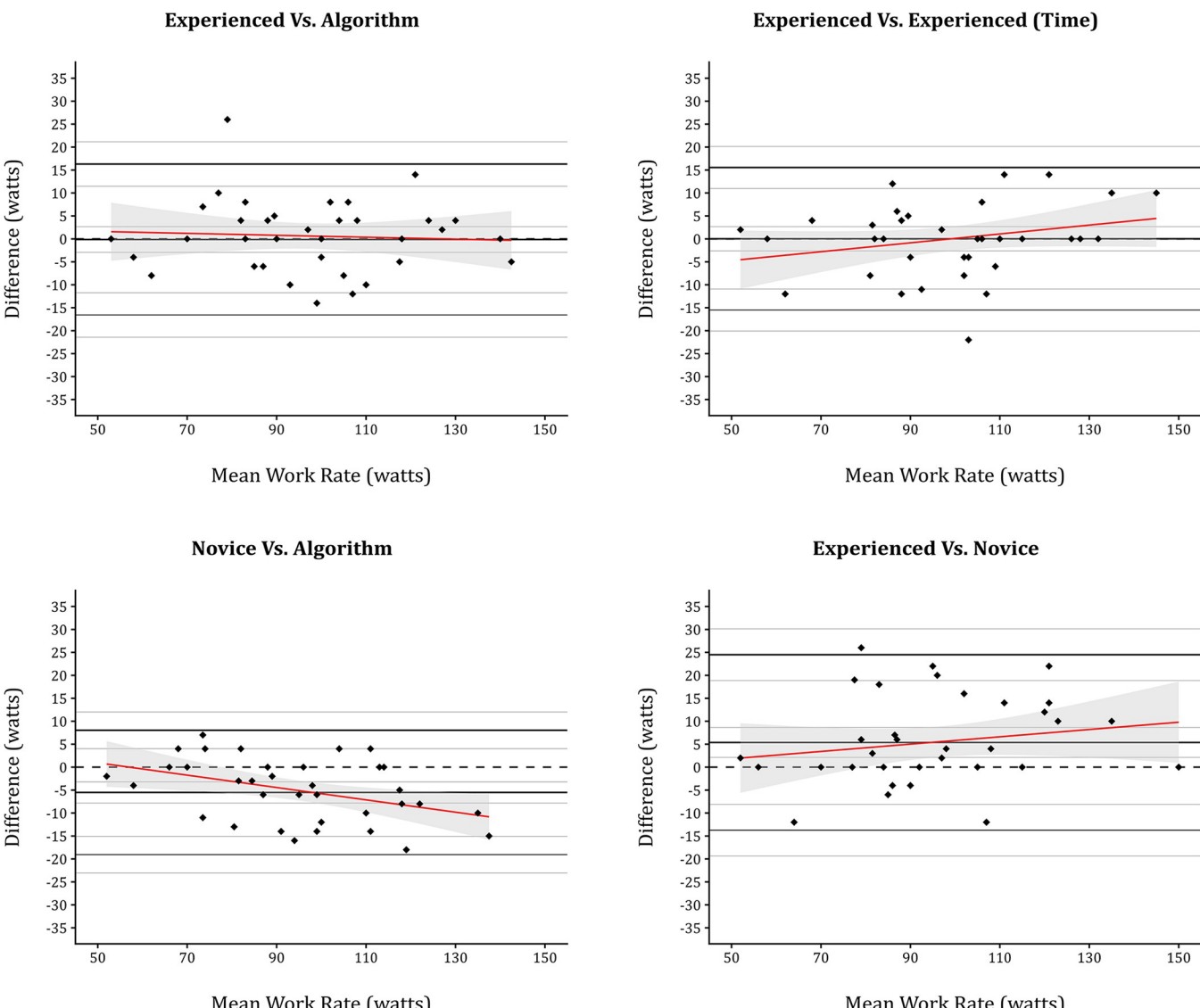

**Fig 3. Bland-Altman plots comparing agreement across subjective and objective inflection identification for work rate.**

experienced examiner. Furthermore, the mean bias when comparing the novice against the algorithm and experienced examiner was consistently different from zero. Indeed, in all comparisons involving the novice examiner there was a statistically significant level of systematic bias (Figs 2 and 3). This systematic bias was compounded by significant proportional bias for comparisons versus the algorithm for work rate representing a statistically significant degree of both systematic and proportional bias. Indeed, all other comparison involving the novice examiner yielded statistically significant systematic bias.

## Discussion

The primary aim of the study was to determine the inter- and intra-observer reliability and agreement of a subjective quantification method for identifying inflections in $O_2$Pulse during cardiopulmonary exercise testing (CPET). Secondly, we sought to establish a suitable objective algorithm as an alternative methodology.

**Table 2. Reliability and agreement analysis for inflection thresholds in heart rate and work rate.**

| | Work Rate | | | |
|---|---|---|---|---|
| Statistic | Algorithm Vs. Experienced | Algorithm Vs. Novice | Experienced Vs. Novice | Experienced Vs. Experienced (Time) |
| ICC | 0.97 (0.95–0.99) | 0.97 (0.86–0.99) | 0.95 (0.87–0.98) | 0.95 (0.90–0.97) |
| Lower LOA (95% CI) | -16.57 (-21.42 - -11.73) | -19.06 (-23.05 - -15.07) | -13.74 (-19.37 - -8.11) | -15.49 (-20.07 - -10.92) |
| Upper LOA (95% CI) | 16.30 (11.46–21.15) | 8.03 (4.04–12.02) | 24.50 (18.86–30.13) | 15.55 (10.98–20.13) |
| Mean Bias (95% CI) | -0.14 (-2.93–2.66) | -5.51 (-7.82 - -3.21) | 5.38 (2.13–8.63) | 0.03 (-2.61–2.69) |
| Systematic bias | $P = 0.92$ | $P < 0.0001$**** | $P = 0.002$** | $P = 0.98$ |
| Proportional bias | $P = 0.11$ | $P = 0.002$** | $P = 0.48$ | $P = 0.15$ |
| | Heart Rate | | | |
| Statistic | Algorithm Vs. Experienced | Algorithm Vs. Novice | Experienced Vs. Novice | Experienced Vs. Experienced (Time) |
| ICC | 0.98 (0.96–0.99) | 0.92 (0.82–0.96) | 0.91 (0.81–0.95) | 0.95 (0.91–0.98) |
| Lower LOA (95% CI) | -10.68 (-13.68 - -7.67) | -21.10 (-26.13 - -16.07) | -14.77 (-20.17 - -9.38) | -10.34 (-13.25 - -7.43) |
| Upper LOA (95% CI) | 9.70 (6.70–12.71) | 13.04 (8.01–18.07) | 21.85 (16.46–27.25) | 9.42 (6.51–12.33) |
| Mean Bias (95% CI) | -0.49 (-2.22–1.25) | -4.03 (-6.93 - -1.12) | 3.54 (0.43–6.66) | 0.46 (-2.14–1.22) |
| Systematic bias | $P = 0.57$ | $P = 0.008$*** | $P = 0.027$* | $P = 0.58$ |
| Proportional bias | $P = 0.41$ | $P = 0.83$ | $P = 0.78$ | $P = 0.77$ |

ICC = Intraclass Correlation Coefficient; LOA = limits of agreement; CI = confidence interval

\* = $P \leq 0.05$

\*\* = $P \leq 0.01$

\*\*\* = $P \leq 0.001$

\*\*\*\* = $P \leq 0.0001$

Before attempting to quantify the threshold of inflection using the proposed algorithm it was necessary to determine whether it could differentiate between normal and abnormal data. The results of the present study indicate that the algorithm can differentiate between data, providing excellent agreement when compared with both experienced and novice examiners. Previous research by de Lorenzo et al and Chuang et al have reported levels of inter-rater reliability of between κC = 0.65 and κC = 0.69 [10, 15] when categorising O$_2$Pulse files, the value reported in the present study, however, are substantially higher at κF = 0.89 (0.83–0.93). There may be several reasons for this disparity. Firstly, the aforementioned studies applied Cohens' Kappa, as they were interested in the agreement of two examiners, whilst we applied Fleiss' Kappa to account for three 'examiners'. Although the additional 'examiner' in this analysis introduces the possibility of greater variability, examiners were only required to score across two categories, that is "Yes" or "No" to indicate the perceived presence or absence of an inflection or plateau. In contrast, the study by de Lorenzo and colleagues [10] required two experienced examiners to place files into one of four categories ('normal', 'probably normal', 'probably abnormal', ' definitely abnormal'), resulting in double the variation in choice afforded in the present study. Similarly, Chuang and co-workers [14] placed an algorithm against the consensus of two human examiners, providing three choices for categorisation ('increasing', 'plateau', and 'decreasing').

The intra- and inter-observer reliability for subjective threshold quantification was assessed by two formats of ICC (2,1; 3,1). The analysis showed excellent reliability in both the intra- (r = 0.95) and inter-rater (r = 0.91–0.95) comparisons, irrespective of the unit of measurement (watts; bpm). As this is a novel methodology, there is no prior data with which to make comparison. However, the technique itself is reminiscent of the modified V-slope method [17], and thus comparisons with its reliability are perhaps justified. In this context, the subjective threshold quantification performs comparably well, as the modified V-slope reported intra-

rater reliability of r = 0.83 when measured using treadmill, and r = 0.96 on cycle ergometry. Similarly, the inter-rater reliability is reported to be r = 0.98 (treadmill) and r = 0.91 (cycle ergometer). However, the mean bias for inter-rater comparisons of both work rate and heart rate was consistently different, as is evident from the significant levels of systematic bias (work rate $P$ = 0.002; heart rate $P$ = 0.027). This presents a substantial hurdle if inflection thresholds for $O_2$Pulse are to be used in a similar way to ventilatory thresholds, for example, to quantify health status and prescribe exercise. For example, the same participant, given the same CPET could be prescribed wholly different exercise intensities by two investigators. This difference appears to be mitigated somewhat if the same examiner were to receive the same CPET, as is reflected by the MDC (15 watts; 10 bpm) and consistent mean bias values (0.03 watts; 0.46 bpm) recorded for the experienced examiner.

The normal progression of $O_2$Pulse during CPET is linear or slightly curvilinear in nature, as stroke volume increase to peak exercise [23]. In such cases, the filtered and smoothed $O_2$Pulse should peak in the latter stages of incremental exercise testing, especially when $\leq$ 90% predicted $VO_{2peak}$ has been achieved. Based on these logical assumptions the proposed algorithm identifies when peak values occur $\geq$ 90 seconds prior to the end of exercise and labels them as points of inflection. The proposed algorithm would inherently have zero intra rater reliability and zero MDC, assuming it were executed as intended. The inter-rater reliability of the algorithm when compared to both experienced and novice examiners was excellent (r = 0.92–0.98). However, as was seen with the experienced and novice examiner comparison there was a significant level of systematic bias when the algorithm and novice operator was compared. As bias was not present in the experienced versus algorithm comparison, it is perhaps more suggestive of a difference in interpretation that stems from level of experience. Furthermore, the limits of agreement and mean bias for algorithm and experienced examiner comparisons were almost identical to those observed in intra-examiner comparisons. Thus, the algorithm could theoretically replace the experienced examiner and eradicate intra-observer variability.

In real-world applications, the experience of clinicians and rehabilitators is wide ranging, thus, the adoption of an objective means of quantification is likely preferable to ensure equality of outcome for patients. For example, guidelines presented by the American College of Sports Medicine (ACSM) [24] suggest exercise intensities for CR to be below the ischaemic threshold (<10 beats), or a threshold that elicits the onset of angina symptoms. When following this guidance it would be preferable to know that, given the same baseline CPET, patients would be receive the same intensity recommendations irrespective of the site they test at or the examiner who reviews their results.

## Limitations

The study is limited by both the small sample of examiners and the accompanying heterogeneous level of experience. Moreover, as there was no invasive ischaemic assessment, inflections in $O_2$Pulse are not guaranteed to align with the onset of myocardial ischaemia. There are two avenues of enquiry for future research to pursue, firstly the algorithm could be used in conjunction with myocardial scintigraphy in an effort to corroborate the ischaemic threshold. Secondly, a larger sample of examiners with diverse levels of training and experience could be used to further establish the agreement of subjective threshold quantification and algorithm performance.

In conclusion, this study provides evidence for the reliability of both subjective and novel objective methods for identifying inflections in $O_2$Pulse during CPET. These findings have important implications for the use of CPET in clinical populations, and suggest that further research into the clinical significance of $O_2$Pulse inflections is warranted.

## Author Contributions

**Conceptualization:** Thomas Nickolay.

**Data curation:** Gordon McGregor, Richard Powell, Brian Begg, Stefan Birkett, Simon Nichols, Stuart Ennis, Prithwish Banerjee, Rob Shave, Lee Ingle.

**Formal analysis:** Thomas Nickolay, James Metcalfe.

**Investigation:** Gordon McGregor, Lee Ingle.

**Methodology:** Thomas Nickolay.

**Resources:** Gordon McGregor.

**Visualization:** Thomas Nickolay.

**Writing – original draft:** Thomas Nickolay.

**Writing – review & editing:** Thomas Nickolay, Gordon McGregor, Richard Powell, Brian Begg, Stefan Birkett, Simon Nichols, Stuart Ennis, Prithwish Banerjee, Rob Shave, James Metcalfe, Angela Hoye, Lee Ingle.

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
