## [Decision Letter · Decision Letter 0]

2 Feb 2024

PONE-D-23-36999Inter- and Intra-Observer Reliability and Agreement of O2Pulse Inflection during Cardiopulmonary Exercise Testing: A Comparison of Subjective and Novel Objective MethodologyPLOS ONE

Dear Dr. Nickolay,

Thank you for submitting your manuscript to PLOS ONE. After careful consideration, we feel that it has merit but does not fully meet PLOS ONE’s publication criteria as it currently stands. Therefore, we invite you to submit a revised version of the manuscript that addresses the points raised during the review process.

**ACADEMIC EDITOR: Dear Author,****Please attend to the comments provided by the reviewer/s and make the necessary changes.** The decision of this manuscript is justified based on PLOS ONE’s publication criteria and not on its novelty or perceived impact.

We look forward to receiving your revised manuscript.

Kind regards,

Zulkarnain Jaafar

Academic Editor

PLOS ONE

Journal Requirements:

Reviewers' comments:

Reviewer's Responses to Questions

**Comments to the Author**

1. Is the manuscript technically sound, and do the data support the conclusions?

Reviewer #1: Yes

Reviewer #2: Yes

Reviewer #3: Yes

2. Has the statistical analysis been performed appropriately and rigorously? 

Reviewer #1: Yes

Reviewer #2: Yes

Reviewer #3: Yes

3. Have the authors made all data underlying the findings in their manuscript fully available?

Reviewer #1: Yes

Reviewer #2: Yes

Reviewer #3: Yes

4. Is the manuscript presented in an intelligible fashion and written in standard English?

Reviewer #1: Yes

Reviewer #2: Yes

Reviewer #3: Yes

5. Review Comments to the Author

Reviewer #1: Generally, this article is clear, well-written and concise. The introduction is relevant and theory based. The authors provided enough and sufficient information about the previous studies' results for readers to understand the rationale and aim of this study. The methods are generally appropriate. Overall, the results are clear and compelling. Moreover, this is a high quality manuscript that has implications for the theoretical basis, diagnosis and identification of inflections in O2Pulse during CPET

Reviewer #2: In my opinion, the article is complete and does not need corrections. This article can be one of the good articles of the magazine.

The only suggestion I have is that they prepare a schematic figure for the methodology..

Reviewer #3: In the present study, the authors investigated the inter-and intra-observer reliability and agreement of O2 pulse inflection and compared subjective measure of O2 pulse inflection with a novel objective method. The study is interesting in nature and the manuscript is well organized. I put some comments on the manuscripts for authors’ consideration.

1. Line 86-87; according to the Fick principle, O2 pulse depends on both SV and Ca-vO2.

2. Line 151-153; Did the authors used any criteria for including the HIIT or MISS trial data? I recommend the author to provide more information in this regard. For instance, why just the tests performed on cycle ergometer were included?

3. Line 164; I recommend the author to provide clear definition or specific criteria for defining “experienced” and “novice” examiner in this study.

4. Line 169; What was the exact time between the first and second observation of the experienced examiner?

5. From the methodological point of view, despite the fact that it was mentioned in the limitation section, when the main goal of this study was to establish a novel objective method by comparing it with a common subjective method, using a novice examiner and basing the conclusion on that might raise some concerns. Please elaborate more on that.

6. PLOS authors have the option to publish the peer review history of their article (what does this mean?). If published, this will include your full peer review and any attached files.

Reviewer #1: No

Reviewer #2: No

Reviewer #3: **Yes: **Ehsan Amiri

---

## [Author Response · Author response to Decision Letter 0]

8 Feb 2024

Response to Reviewers

“Reviewer #1: Generally, this article is clear, well-written and concise. The introduction is relevant and theory based. The authors provided enough and sufficient information about the previous studies' results for readers to understand the rationale and aim of this study. The methods are generally appropriate. Overall, the results are clear and compelling. Moreover, this is a high quality manuscript that has implications for the theoretical basis, diagnosis and identification of inflections in O2Pulse during CPET”

Thank you for your comments.

“Reviewer #2: In my opinion, the article is complete and does not need corrections. This article can be one of the good articles of the magazine.

The only suggestion I have is that they prepare a schematic figure for the methodology..”

Thank you for the comments, we are unclear what was meant by a ‘schematic figure”, if the reviewer was indicating a flowchart indicating either the research design or recruitment through the study we would disagree given the simple nature of the study design and relatively small sample size. Each step is covered in detail in the methods and the manuscript already contains three figures and two tables.

“Reviewer #3: In the present study, the authors investigated the inter-and intra-observer reliability and agreement of O2 pulse inflection and compared subjective measure of O2 pulse inflection with a novel objective method. The study is interesting in nature and the manuscript is well organized. I put some comments on the manuscripts for authors’ consideration.

1. Line 86-87; according to the Fick principle, O2 pulse depends on both SV and Ca-vO2.

Thank you for the comments, this terminology is widely accepted in the field and appears in multiple sources cited throughout the manuscript. We have altered the sentence to read – “In principle, O2Pulse reflects left ventricular stroke volume (SV) (and arteriovenous oxygen difference) (1,4).”

2. Line 151-153; Did the authors used any criteria for including the HIIT or MISS trial data? I recommend the author to provide more information in this regard. For instance, why just the tests performed on cycle ergometer were included?

All available data were included, only those missing data relating to oxygen consumption or heart rate were not considered. The protocol dictated exercise be performed on cycle ergometer and therefore there is no treadmill data available. Cycle ergometry was principally included as workload/power output could be controlled and used to fine-tune progressive overload during the training programme. This is explained in the HITT or MISS UK protocol paper published in 2016. 

3. Line 164; I recommend the author to provide clear definition or specific criteria for defining “experienced” and “novice” examiner in this study.

We have adjusted the manuscript to read “However, one examiner had substantially more experience with O2Pulse morphology (>6 years) and inflections. This examiner is subsequently referred to as ‘experienced’ whilst the other is termed ‘novice’ (<1 year).”

4. Line 169; What was the exact time between the first and second observation of the experienced examiner?

The times varied due to the volume of data and the time commitment involved. We have altered the manuscript to read “The ‘experienced’ examiner revisited the ‘yes’ curves at a later date (7- 14 days) to re-quantify the inflection threshold.”

5. From the methodological point of view, despite the fact that it was mentioned in the limitation section, when the main goal of this study was to establish a novel objective method by comparing it with a common subjective method, using a novice examiner and basing the conclusion on that might raise some concerns. Please elaborate more on that.”

In practice novice examiners are involved in data interpretation alongside more experienced colleagues. Thus, including the novice examiners allows for more real-world practical interpretations to be made. Furthermore, the experienced examiner is the one for whom intra-rater comparisons were made.

---

## [Decision Letter · Decision Letter 1]

12 Feb 2024

Inter- and Intra-Observer Reliability and Agreement of O2Pulse Inflection during Cardiopulmonary Exercise Testing: A Comparison of Subjective and Novel Objective Methodology

PONE-D-23-36999R1

Dear Dr. Nickolay,

We’re pleased to inform you that your manuscript has been judged scientifically suitable for publication and will be formally accepted for publication once it meets all outstanding technical requirements.

Kind regards,

Zulkarnain Jaafar

Academic Editor

PLOS ONE

Additional Editor Comments (optional):

Reviewers' comments:

Reviewer's Responses to Questions

**Comments to the Author**

1. If the authors have adequately addressed your comments raised in a previous round of review and you feel that this manuscript is now acceptable for publication, you may indicate that here to bypass the “Comments to the Author” section, enter your conflict of interest statement in the “Confidential to Editor” section, and submit your "Accept" recommendation.

Reviewer #3: All comments have been addressed

2. Is the manuscript technically sound, and do the data support the conclusions?

Reviewer #3: Yes

3. Has the statistical analysis been performed appropriately and rigorously? 

Reviewer #3: Yes

4. Have the authors made all data underlying the findings in their manuscript fully available?

Reviewer #3: Yes

5. Is the manuscript presented in an intelligible fashion and written in standard English?

Reviewer #3: Yes

6. Review Comments to the Author

Reviewer #3: The authors have addressed all my comments, and the manuscript is now appropriate for publication. I have no further comments.

7. PLOS authors have the option to publish the peer review history of their article (what does this mean?). If published, this will include your full peer review and any attached files.

Reviewer #3: **Yes: **Ehsan Amiri

---

## [Editor Report · Acceptance letter]

28 Feb 2024

PONE-D-23-36999R1 

PLOS ONE

Dear Dr. Nickolay, 

I'm pleased to inform you that your manuscript has been deemed suitable for publication in PLOS ONE. Congratulations! Your manuscript is now being handed over to our production team.

Kind regards, 

on behalf of

Dr. Zulkarnain Jaafar 

Academic Editor

PLOS ONE